# Immune Profile Determines Response to Vaccination against COVID-19 in Kidney Transplant Recipients

**DOI:** 10.3390/vaccines11101583

**Published:** 2023-10-11

**Authors:** Stamatia Stai, Asimina Fylaktou, Efstratios Kasimatis, Aliki Xochelli, Georgios Lioulios, Vasiliki Nikolaidou, Anastasia Papadopoulou, Grigorios Myserlis, Artemis Maria Iosifidou, Myrto Aikaterini Iosifidou, Aikaterini Papagianni, Evangelia Yannaki, Georgios Tsoulfas, Maria Stangou

**Affiliations:** 1School of Medicine, Aristotle University of Thessaloniki, 45642 Thessaloniki, Greece; staimatina@yahoo.gr (S.S.); frasci@outlook.com.gr (E.K.); pter43@yahoo.gr (G.L.); grmiserlis@gmail.com (G.M.); ios.artemis.2000@gmail.com (A.M.I.); ios.myrtv.2002@gmail.com (M.A.I.); aikpapag@auth.gr (A.P.); tsoulfasg@auth.gr (G.T.); 2Department of Nephrology, Hippokration Hospital, 54642 Thessaloniki, Greece; 3Department of Immunology, National Histocompatibility Center, Hippokration General Hospital, 54642 Thessaloniki, Greece; fylaktoumina@gmail.com (A.F.); aliki.xochelli@gmail.com (A.X.); basoniko@hotmail.com (V.N.); 4Hematology Department, Hematopoietic Cell Transplantation Unit, Gene and Cell Therapy Center, “George Papanikolaou” Hospital, 57010 Thessaloniki, Greece; apapadopoulou.gpapanikolaou@n3.syzefxis.gov.gr (A.P.); eyannaki@uw.edu (E.Y.); 5Department of Transplant Surgery, Hippokration General Hospital, 54642 Thessaloniki, Greece

**Keywords:** kidney transplantation, COVID-19 vaccination, lymphocytes, protective humoral response, cellular response, ELISpot

## Abstract

Background and Aim: Immune status profile can predict response to vaccination, while lymphocyte phenotypic alterations represent its effectiveness. We prospectively evaluated these parameters in kidney transplant recipients (KTRs) regarding Tozinameran (BNT162b2) vaccination. Method: In this prospective monocenter observational study, 39 adult KTRs, on stable immunosuppression, naïve to COVID-19, with no protective humoral response after two Tozinameran doses, received the third vaccination dose, and, based on their immunity activation, they were classified as responders or non-responders. Humoral and cellular immunities were assessed at predefined time points (T_0_: 48 h before the first, T_1_: 48 h prior to the third and T_2_: three weeks after the third dose). Results: Responders, compared to non-responders, had a higher total and transitional B-lymphocyte count at baseline (96.5 (93) vs. 51 (52)cells/μL, *p*: 0.045 and 9 (17) vs. 1 (2)cells/μL, *p*: 0.031, respectively). In the responder group, there was a significant increase, from T_0_ to T_1,_ in the concentrations of activated CD4+ (from 6.5 (4) to 10.08 (11)cells/μL, *p:* 0.001) and CD8+ (from 8 (19) to 14.76 (16)cells/μL, *p*: 0.004) and a drop in CD3+PD1+ T-cells (from 130 (121) to 30.44 (25)cells/μL, *p*: 0.001), while naïve and transitional B-cells increased from T_1_ to T_2_ (from 57.55 (66) to 1149.3 (680)cells/μL, *p* < 0.001 and from 1.4 (3) to 17.5 (21)cells/μL, *p*: 0.003). The percentages of memory and marginal zone B-lymphocytes, and activated CD4+, CD8+ and natural killer (NK) T-cells significantly increased, while those of naïve B-cells and CD3+PD1+ T-cells reduced from T_0_ to T_1_. Conclusions: Responders and non-responders to the third BNT162b2 dose demonstrated distinct initial immune cell profiles and changes in cellular subpopulation composition following vaccination.

## 1. Introduction

Coronavirus disease 2019 (COVID-19) presents with a wide spectrum of clinical manifestations, ranging from asymptomatic disease to life-threatening respiratory involvement and various extrapulmonary signs and symptoms [1,2,3,4,5]. Risk factors predisposing to adverse outcomes comprise old age; male gender; and several comorbidities, including kidney disease and immunodeficiencies [6]. Being significantly immunocompromised mainly due to the use of immunosuppressants and presenting with a pleiad of coexisting medical conditions, kidney transplant recipients (KTRs) have a higher incidence of COVID-19 infection and are more prone to the development of severe disease complications [7,8].

Over the first two years of the pandemic, a considerable number of different vaccines against SARS-CoV-2 were developed, including messenger RNA (mRNA), inactivated virus, viral vector and recombinant protein-based platforms [8,9]. Tozinameran (BNT162b2), a nucleoside-modified mRNA vaccine created by Pfizer-BioNTech, has been the most widely used one among Greek KTRs. The enclosed mRNA molecule, incorporated into lipid nanoparticle carriers, is responsible for encoding the full-length SARS-CoV-2 spike (S) protein. Exposure of the produced antigen (Ag) on the cell surface is capable of eliciting robust cluster of differentiation (CD)4+- and CD8+-mediated immune responses [10,11,12,13].

There is no doubt that vaccines constitute a crucial weapon in our attempt to combat complex infections. Given that, a lot of research has been conducted in order to assess to what extent an individual’s pre-vaccination “baseline” molecular and cellular state can be useful in gaining a better understanding of the mechanisms determining the vaccine-mediated response, as well as in predicting its magnitude [14]. In the same context, post-vaccination changes in cellular subset configurations can also be indicative of the immunization procedure’s effectiveness [15]. As anticipated, many investigators have recently focused on SARS-CoV-2, attempting to shed light on the potential connection between vaccination-induced antibody (Ab) production and initial concentrations, as well as dynamic alterations in the numbers of several T- and B-cell subsets [15,16].

KTRs consist of an immunologically distinct group of patients and have been extensively studied with regards to their cellular subpopulation composition. Indeed, compared to healthy individuals, they tend to have a lower thymic output, as well as higher concentrations of aged and exhausted lymphocytes, rendering their vaccination-mediated immunity activation less robust [8]. For this reason, their response to vaccination is of particular interest in order to apply different protocols adapted to the needs of immunocompromised patients. Furthermore, in this particular patient cohort, we need to assess the parameters and situations that will probably influence their response to any vaccination. In the present research, we aimed to assess the impact of the patients’ initial immune cell subpopulation composition on their ability to respond to BNT162b2, as well as the effect of potential vaccination-elicited immunization on the numbers and proportions of various lymphocyte subsets. It is important to mention that we attempted to achieve a more thorough approach, taking into account both cellular and humoral response development in our patients.

Ideally, data emerging from our research, adding to the already existing knowledge, could aid in the search for new perspectives with regards to the adoption of more personalized vaccination schemes, thus promoting the implementation of more viable immunization strategies, as well as an increase in patient compliance in the long term.

## 2. Materials and Methods

### 2.1. Patients

Our sample consisted of KTRs who were being followed up in the outpatient clinics of the Department of Nephrology and the Department of Transplant Surgery of the Aristotle University of Thessaloniki. All of them were thoroughly informed about the purpose, schedule and requirements of the study, and they signed a consent form prior to their enrolment.

The study was conducted in accordance with the Declaration of Helsinki and approved by the Institutional Review Board of Hippokration General Hospital, Thessaloniki, Greece, Approval Number 675/21. All patients were informed before entering the study, and they signed a consent form, 676/21.

#### 2.1.1. Inclusion Criteria

The participants were adults (≥18 years old) who had received a kidney graft at least 3 months before their recruitment. They were all being followed up in our outpatient clinics and were treated with stable dosages of immunosuppressives (a triple combination of corticosteroids, calcineurin inhibitors and mycophenolate mofetil) during the whole study period.

Patients were required to have no history of COVID-19 infection before vaccination and proven undetectable immunity against the SARS-CoV-2 virus. The latter was defined as the absence of protective humoral immunity (PHI) at time points T_0_ and T_s_.

#### 2.1.2. Exclusion Criteria

Individuals with comorbidities (including systemic diseases, solid tumors or hematopoietic cell malignancies) or with recent (during the past two years) administration of chemotherapeutic regimens or rituximab could not be enrolled in our study. Moreover, those with acute cellular or humoral rejection events (during the semester preceding their recruitment) or recent history of microbial or viral infection (during the last trimester) were also excluded.

### 2.2. Schedule of the Study

Patients’ immune status, including immunity against COVID-19 (PHI and specific T-cell immunity), along with immune cell phenotypes, was assessed at predefined time points following vaccination. Definitions regarding the patients’ immune status are given below.

An analytical schematic diagram of the study schedule is depicted in Figure 1.

Vaccination was carried out in accordance with the guidelines of the Greek National Vaccination Committee: the second dose was administered 3 weeks following the first (at week 3), and the third dose was administered 16 weeks after the second, at week 19.

During the vaccination period, we assessed several aspects of the patients’ immune status at 4 predefined time points, as depicted in Figure 1: 48 h before the first BNT162b2 dose (T_0_), 3 weeks after the second dose (T_s_, week 6), 48 h before the third dose (T_1_, week 19) and three weeks after the third dose (T_2_, week 22).

Humoral immunity was assessed at time point T_s_, and, accordingly, patients were classified as PHI (+) or PHI (−). Further evaluation was performed only in the group of PHI (−) patients, who were considered primary non-responders. More specifically, we examined the immune response that they developed following repeated vaccination doses while also taking into consideration their initial immune profile as a potential predictor of the latter.

Patients were separated into responders and non-responders based on their immunity status against SARS-CoV-2 at the end of the study (at time point T_2_). More precisely, those who had developed PHI and/or specific T-cell immunity by T_2_ were regarded as responders, while those who failed to develop either of the two were considered non-responders.

### 2.3. Definitions

Parameters used to estimate patients’ immune status were as follows:

Immunity against COVID-19, the assessment of which was based on the presence of the following:(i)Protective Humoral Immunity (PHI) against COVID-19, evaluated with regards to the serum concentrations of anti-SARS-CoV-2-neutralizing antibodies (Nabs), measured with the chemiluminescence immunoassay technique. Nab levels ≥ 0.3 AU/mL were regarded as positive, and patients were regarded as PHI (+).(ii)Specific T-cell immunity, detected with the utilization of an enzyme-linked immunosorbent spot (ELISpot) test. ELISpot test values ≥ 30 spot-forming cells (SFC)/5 × 10^5^ peripheral blood mononuclear cells (PBMCs) were considered positive.

Their repertoire of lymphocyte and mononuclear cell phenotypes, as described below.

The final response to vaccination was based on the presence of PHI and/or specific T-cell immunity, and patients were subsequently classified as responders or non-responders.

### 2.4. Laboratory Methods

#### 2.4.1. Flow Cytometry

Heparinized blood samples were selected from our patients at all examined time points (T_0_, T_1_ and T_2_) in order to assess the concentrations and percentages of several immune cell subpopulations. The surface molecules that we detected with the use of fluorescent-conjugated Abs, the clone numbers representing the cell lines by which they were produced and the fluorochromes are presented at Table 1.

The flow cytometer instrument that we utilized was the Navios Flow Cytometer, Beckman Coulter, Brea, CA, USA. All examined cell subsets and their markers are presented below:

White blood cells (WBC) (CD45+)

B-lymphocytes (CD45+CD19+): 

  -Memory B-cells (CD27+CD38^low^) 

  -Naïve B-cells (IgD+CD27−) 

  -Plasmablasts (CD27+CD38^high^), which are separated into: 

   i) Class-switched (IgD+/− IgM+) 

   ii) Non-class-switched (IgD− IgM−)

  -Transitional B-cells (CD24+CD38^high^)

  -Marginal zone (MZ) B-cells (IgD+CD27−)

T-lymphocytes (CD45+CD3+): 

  -T-helper (Th) cells (CD4+)

   Activated Th cells (CD38+ HLA-DR+) 

  -Cytotoxic T-cells (CD8+)

   Activated cytotoxic T-cells (CD38+ HLA-DR+)

Monocytes (CD45+CD14+)

  -activated monocytes (CD45+CD14+CD38+HLA-DR+)

Natural Killer (ΝΚ) cells (CD45+CD3−CD16+CD56+), 

NK-like T-lymphocytes (NKT-cells) (CD45+CD3+CD56+), 

  activated NKT (CD45+CD3+CD56+CD27+)

Exhausted T-lymphocytes (CD45+CD3+PD1+)

The gating strategy for the above subpopulations is described in Appendix A.

#### 2.4.2. Chemiluminescence Immunoassay Technique (CLIA)

For the detection of anti-SARS-CoV-2 Nabs, we utilized a competitive chemiluminescence immunoassay (Maglumi™ 2000 Plus-New Industries Biomedical Engineering Co., Ltd. (Snibe), Shenzhen, China). More specifically, we mixed and intubated patient serum (containing Nabs) with a buffer, magnetic microbeads covered with angiotensin-converting enzyme 2 (ACE-2) Ags and Amino-Butyl-Ethyl-Isoluminol (ABEI)-labeled recombinant SARS-CoV-2 S-receptor binding domain (RBD) Ags. Nabs, which were present in the sample, compete with ACE-2 Ags for binding labeled SARS-CoV-2 S-RBD Ags. After sedimentation (with the application of a magnetic field) and the induction of a chemiluminescence reaction, a light signal (measured in relative light units (RLU)) inversely proportional to the serum NAb concentration was produced and detected.

#### 2.4.3. Enzyme-Linked Immunosorbent Spot Test (ELISpot)

Specific T-cell immunity status was assessed with the use of ELISpot. We isolated peripheral blood mononuclear cells and stimulated them with a mix of overlapping fifteen-mer fragments of the full-length S viral protein (jpt peptide Technologies). We subsequently measured the number of spot-forming (i.e., interferon gamma (IFN-γ)-secreting) cells with an Eli.Scan ELISpot scanner (A.EL.VIS) utilizing Eli. Analyse software V6.2.SFC. SARS-CoV-2 spike-specific T-cells are expressed as SFC per input cells, and values 30 ≥ SFC/5 × 10^5^ PBMC were considered positive.

### 2.5. Statistical Analysis

A statistical analysis was performed with IBM SPSS 26.0 (SPSS Inc., Chicago, IL, USA). Results with a *p*-value < 0.05 were regarded as significant. We utilized the Shapiro–Wilk test in order to identify the presence of a normal distribution in all quantitative variables. Since all of them displayed a non-normal distribution, the median value (MED) and interquartile range (IQR) were the preferred measures of central tendency. A comparison among more than two median values was carried out with the Friedman test. Pairwise comparisons for all possible relevant combinations were performed with the Wilcoxon test, and the *p* values were subsequently adjusted with the use of Bonferroni correction. The median values of two independent groups were compared with the Mann–Whitney U test.

## 3. Results

### 3.1. Differences between Responders and Non-Responders

From an initial cohort of 56 KTRs, naïve to SARS-CoV-2 infection and with undetectable anti-SARS-CoV-2 Ab concentrations at T_0_, we excluded those who managed to develop PHI at T_s_, and, thus, our final sample at T_s_ comprised 39 PHI (−) patients.

At T_2_, 34 out of 39 participants had developed PHI and/or anti-SARS-CoV-2-specific cellular immunity and were classified as responders. The baseline clinical parameters of the responders and non-responders are presented at Table 2. Interestingly, no statistically significant differences were observed in age, calcineurin inhibitor levels or transplantation and dialysis vintage.

The differences in the two examined groups regarding cellular subpopulations are displayed in Table 3. The responders demonstrated an initial immune cell profile that was slightly distinct from that of the non-responders, with this being reflected in differences in cell subpopulation concentrations. More specifically, prior to the initiation of the vaccination schedule, they had a higher total B-lymphocyte count (96.5 (93) cells/μL vs. 51 (52) cells/μL in the responders and non-responders, respectively, *p*: 0.045). This finding could mainly be attributed to the greater numbers of transitional B-cells observed among those individuals (9 (17) cells/μL in responders vs. 1 (2) cells/μL in non-responders, *p*: 0.031), while the differences regarding the rest of the B-lymphocyte subsets, as well as the T-cell compartment, were not notable.

### 3.2. Changes in Immune Cell Phenotype during Vaccination

#### 3.2.1. Responders

The recorded fluctuations in cell concentrations between blood samplings were more prominent in the responder group, where we observed a remarkable rise in activated CD4+ and CD8+, as well as a drop in CD3+PD1+ T-cell numbers from T_0_ to T_1_ (at T_0_ vs. T_1_: 6.5 (4) vs. 10.08 (11) cells/μL, *p*: 0.001, 8 (19) vs. 14.76 (16) cells/μL, *p*: 0.004 and 130 (121) vs. 30.44 (25) cells/μL, *p*: 0.001, respectively).

Additionally, these patients presented an elevation in naïve B-lymphocytes (which was actually prominent only at T_2_: naïve B-cell concentrations: 57.55 (66) at T_1_ vs. 1149.3 (680) cells/μL at T_2_, *p* < 0.001) with a concurrent increase in transitional B-cells and a reduction in CD4+ T-cells from T_1_ to T_2_ (at T_1_ vs. T_2_: 1.4 (3) vs. 17.5 (21) cells/μL, *p*: 0.003 and 10.08 (11) vs. 5.29 (5) cells/μL, *p*: 0.042, respectively).

The above-mentioned fluctuations in immune cell concentrations are analytically presented in Table 4 and Figure 2.

#### 3.2.2. Non-Responders

Among the “non-responders”, the majority of alterations regarding cell concentrations were insignificant. Nevertheless, they demonstrated a naïve B-cell number fluctuation during follow-up that was similar yet less prominent compared to that observed in the responders, with a transient, non-notable drop in their counts at T_1_, followed by a significant ascent at T_2_ (16.87 (32) at T_1_ vs. 867.01 (1279) cells/μL at T_2_, *p*: 0.04) and, interestingly, an increase in total lymphocytes and in transitional B-cells in the same time interval (1101.6 (892) at T_1_ vs. 1554 (1270) cells/μL at T_2_, *p*: 0.034 and at T_1_ and at T_2_, *p*: 0.04) (Table 5).

Although individuals who developed a protective immune response following BNT162b2 administration presented a multitude of changes concerning the percentage compositions of several immune cell subpopulations, this was not the case among those who failed to respond to vaccination. Indeed, from T_0_ to T_1_, in the first group of patients, we noticed a rise in the proportion of memory and MZ B-cells and a reduction in that of naïve B-cells with regards to the B-lymphocyte compartment (at T_0_ vs. T_1_: 25.6 (23.9) vs. 32 (26)%, *p*: 0.001, 10.7 (12.4) vs. 14 (12.4)%, *p*: 0.031 and 74.4 (23.7) vs. 69.7 (25.7)%, *p*: 0.002, respectively) (Figure 3).

As for T-cells, there was an elevation in the percentage of activated CD4+, CD8+ and NKT-, as well as a drop in that of CD3+PD1+ T-cells (at T_0_ vs. T_1_: 0.9 (0.9) vs. 1.2 (1.1)%, *p*: 0.001, 1.9 (3.7) vs. 3.2 (2)%, *p*: 0.017, 1.4 (3) vs. 2.6 (2.8)%, *p*: 0.001 and 7.3 (4.7) vs. 1.8 (1.4)%, *p* < 0.001, respectively). From T_1_ to T_2_, we observed a notable reduction in the activated CD4+ and NKT cell proportions, combined with an increase in the CD3+CD8+ percentage (at T_1_ vs. T_2_: 1.2 (1.1) vs. 0.8 (0.6)%, *p*: 0.031, 2.6 (2.8) vs. 1.9 (2.6)%, *p*: 0.03 and 30.4 (12) vs. 30.7 (13.9)%, *p*: 0.002, respectively) (Table 6).

Changes in the proportions of activated CD4+ and CD8+ T-cells during follow-up are depicted in Figure 4.

On the contrary, in the second group of patients, the only recorded statistically significant alteration regarding cell subpopulation proportions was the reduction in CD3+PD1+ T-cells from T_0_ to T_1_ (4.7 (4.1)% vs. 2.9 (2.1)%, *p*: 0.004) (Table 7).

## 4. Discussion

According to our findings, the presence of higher total and transitional B-lymphocyte concentrations prior to BNT162b2 administration is correlated with greater vaccine responsiveness after the third dose. Several studies have already underlined the impact of patients’ baseline immune cell profile on vaccination effectiveness [14]. As anticipated, the response to primary vaccination mostly depends on the numbers of total or naïve B-lymphocytes, while, when it comes to booster doses, it is contingent on late differentiated B-cell (e.g., Ag-specific memory B- and Ab-secreting plasma cell) concentrations [17]. Transitional B-cells are immature forms of B-lymphocytes, serving as mature B-cell precursors. Their maturation process depends on their Ag exposure and results in the translocation of the produced mature cells into germ centers, where they can differentiate into memory cells and, finally, plasma cells [18]. Consequently, it was not to our surprise that patients who effectively responded to the third dose had demonstrated more elevated concentrations of that particular subpopulation at baseline. Moreover, another study by Stefanski et al. comprising immunosuppressed patients also proved that elevated pre-anti-SARS-CoV-2 vaccination transitional B-cell levels had a positive correlation with vaccination-induced immunity activation [19]. Interestingly, according to other investigators, regarding the rest of the B-lymphocyte compartment components, low numbers of memory B-cells (total, switched or unswitched), often noticed among patients receiving IST, seem to account for ineffective immunization, while impaired somatic hypermutation and class-switching processes can also be responsible for poor responsiveness [17].

In our study, the KTRs who developed a protective immune response against SARS-CoV-2 three weeks after the third BNT162b2 dose exhibited an upward trend in activated CD4+ and CD8+ T-cell numbers from T_0_ to T_1_, as well as a reduction in CD3+PD1+ T-lymphocyte concentrations during the same time interval. Moreover, we observed a decline in activated CD4+ T-cell counts and an increase in naïve and transitional B-cell counts from T_1_ to T_2_.

According to other authors, in naïve individuals, SARS-CoV-2-specific CD4+ T-cell formation can be induced shortly after the first Tozinameran dose and has a functional role in adaptive immunity development (as reflected in the positive correlation observed between pre-second dose T helper cell type 1 (Th1) and circulating T-follicular cell (cTfh) levels and post-second dose CD8+ and NAb concentrations, respectively). On the contrary, the induction of Ag-specific CD8+ T-cells is more gradual. Moreover, it seems that, in contrast to convalescent patients, SARS-CoV-2-naïve individuals can significantly benefit from booster vaccination with regards to T-cell response promotion [20].

Another study by Lioulios et al., performed in hemodialysis (HD) patients, proved that, four months after the second BNT162b2 dose, protective humoral immunity was associated with naïve CD4+, as well as with late differentiated T-cell counts, possibly through the upregulation of Th 17 cells, which promote the formation of plasmablasts and regulatory T-cells [21]. It is known that the production of effector T-cells follows the activation and consequent clonal expansion of naïve forms after Ag recognition [22,23]. Although the current literature has not yet clarified whether effector memory (EM) T-cells derive from effector T-cells or vice versa, it is quite clear that the concentration of late differentiated and EM T-lymphocytes is directly connected to that of activated forms [22,23]. Based on the previously presented data, we can state that the increase in activated CD4+ and CD8+ numbers that we observed in vaccination responders four months after the second dose is in accordance with the findings of Lioulios et al. [21].

As for our results regarding the fluctuations in the B-lymphocyte compartment subpopulations after vaccination, the post-third dose increase that we observed in transitional B-cell concentrations was in accordance with the findings of other investigators, who also noticed an expansion of this cell subset among vaccination responders following antigen stimulation. These findings can perhaps be explained in the context of the vital role that transitional B-lymphocytes play in mature B-cell generation by suppressing the production of proinflammatory cytokines, thus promoting B-cell survival and proliferation, as well as Ab production [24]. On the contrary, the increase in naïve B-cell number from T_1_ to T_2_ was unexpected and quite paradoxical.

Another interesting observation was the gradual drop in PD-1-expressing T-cells, following the administration of the second and third doses (in the time intervals T_0_-T_1_ and T_1_-T_2_, respectively), that we observed in the responder group. PD-1 is a transmembrane protein serving as a major cell exhaustion regulator by mediating adaptive and innate immune response inhibition. It is found in a pleiad of hematopoietic cells, including activated T-lymphocytes. Programmed cell death-1 ligand-1 (PD-L1), constitutively expressed in macrophages, some activated B- and T-lymphocytes, and dendritic and epithelial cells, interacts with PD-1, halting proliferation, impeding cytokine production and, finally, mediating the apoptosis of PD-1-positive cells [25,26,27]. Although the activation of the PD-1/PD-L1 pathway (which has been extensively studied in malignancy) seems to contribute to donor alloantigen tolerance in solid organ transplantation (SOT) [28], there are only a few research works examining its effect on KTRs. Of note, a recent study demonstrated that, after anti-SARS-CoV-2 vaccination, peripheral blood granulocytes and monocytes display an increased expression of PD-L1 (possibly in the context of collateral autoimmunity damage prevention) [29]. Given its key role in PD-1-positive cell apoptosis induction, this finding could partially explain the reduction that we noticed in CD3+PD-1+ cell numbers.

Regarding the alterations in the proportions of the cellular subpopulations constituting the B- and T-lymphocyte compartments in the responder group, from T_0_ to T_1_, we noticed an increase in the percentage of activated, effector and memory forms (more specifically, of memory and MZ B-cells and of activated CD4+, CD8+ and NKT cells) with a concurrent reduction in naïve and exhausted ones. From T_1_ to T_2_, there was a drop in activated CD4+ and NKT cell proportions, as well as a slight rise in the proportion of CD8+ T-lymphocytes.

In accordance with the rationale behind our findings on B-cells, other investigators have also supported the notion that, among COVID-19-convalescent individuals, a second vaccination-mediated exposure to viral Ags can cause an expansion of the Ag-specific memory B-cell pool and a consequent augmentation of the Ab response. On the contrary, according to their results, booster dose administration is not responsible for notable changes in the B-lymphocyte compartment composition. The case is quite similar to that of previously vaccinated patients who present an increase in pre-existing virus-specific memory B-cells after receiving a second vaccination dose [30]. As for MZ B-lymphocytes, it seems that their role is not solely limited to T-independent immune response orchestration. It has been proved that, additionally, after immunization with protein Ags, they are capable of inducing a robust (and sometimes superior to the follicular B-cell-mediated) CD4+ T-cell expansion, promoting the activation of the latter and their transformation into effector cells. Moreover, they can also differentiate into plasma cells and stimulate Th1 cytokine production [31]. To some extent, this can explain why we observed a concurrent increase in MZ B-cells, CD4+ T-cells and Th1-cytokine-producing (i.e., activated CD8+ and activated NKT) cells from T_0_ to T_1_. Regarding the non-anticipated slight drop in the activated CD4+ count and percentage and in the activated NKT cell percentage from T_1_ to T_2_, we can only assume that the expected post-third dose increase in their proportions is likely to occur later after vaccination. The rise in total CD8+ T-lymphocyte percentage during the same time interval could possibly be attributed to the previously mentioned delayed expansion of Ag-specific CD8+ T-cells following vaccination [20].

Finally, we should mention some limitations of our study. Firstly, we examined a rather small number of patients, as our final sample consisted of 39 individuals selected from an initial pool of 56. They were all stable adult patients, naïve to COVID-19, on fixed and unmodified immunosuppressive treatments, with no comorbidities, recent administration of chemotherapeutics, acute rejection events or history of infection. Although the implementation of all the above-mentioned inclusion and exclusion criteria resulted in the creation of a very homogenous group of participants, the generalization of our findings may be unsafe. In the same context, we should underline that the number of included non-responders was disproportionally small compared to that of responders (they comprised only 4 out of the 39 KTRs). Another drawback of our research is the fact that we did not manage to examine any possible correlations between the dosage of the administered immunosuppressive treatment and vaccination responsiveness. Nevertheless, our findings still contain valuable information with regards to the adoption of a more individualized immunization strategy. Given the fact that an unfavorable combination of “technical” difficulties and decreased patient compliance have emerged after the COVID-19 pandemic outbreak [32], such steps towards a more patient-centric and, thus, effective approach can potentially lead to the future achievement of higher immunization rates and, consequently, improved disease control.

## 5. Conclusions

As anticipated, the KTRs’ response to BNT162b2 was influenced by their pre-vaccination immunity status. Moreover, we observed differences between responders and non-responders with regards to alterations in cellular subset numbers and percentages following vaccination. To the best of our knowledge, our study is one of the very few research works thoroughly examining the potential dynamic interaction between cellular subpopulation composition and vaccination responsiveness in this immunologically distinct group of patients. We believe that, together with the rest of the papers published on this special issue, our findings will add important information regarding the particular response of KTRs to vaccination and the parameters that determine this response.

Of course, a lot of additional work needs to be conducted in this direction in order to obtain stronger results that can be generalized more reliably. Nevertheless, we hope that, to some degree, the findings of our study will shed light on the novel pathophysiologic mechanisms implicated in vaccination-induced immunity activation in KTRs. That could potentially offer new future perspectives and help in the selection of more individualized vaccination strategies for groups of immunocompromised patients.

## Figures and Tables

**Figure 1 vaccines-11-01583-f001:**
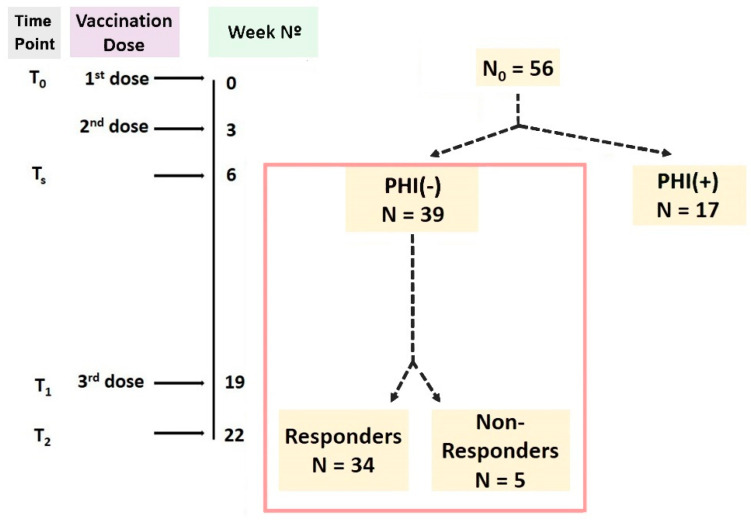
Schedule of the study. The initial pool consisted of 56 KTRs naïve to COVID-19 (as proven by the absence of humoral immunity against SARS-CoV-2 at T_0_). Our final sample (indicated inside the red box) consisted solely of individuals who had failed to develop protective humoral immunity by T_selection_ (T_s_), and they were characterized as PHI (−) patients. These patients were later separated into responders (PHI (+) and/or ELISpot (+), N = 34) and non-responders (PHI (−) and ELISpot (−), N = 5). Utilizing flow cytometry, we evaluated immune cell phenotypes at T_0_, T_1_ and T_2_ and compared their concentrations and percentages between responders and non-responders. Explanation of selected time points: T_0_: baseline—week 0, 48 h prior to the 1st BNT162b2 dose administration; T_s_: 3 weeks after the 2nd BNT162b2 dose administration, week 6; T_1_: 48 h prior to the 3rd BNT162b2 dose administration, week 19; T_2_: 3 weeks after the 3rd BNT162b2 dose administration, week 22.

**Figure 2 vaccines-11-01583-f002:**
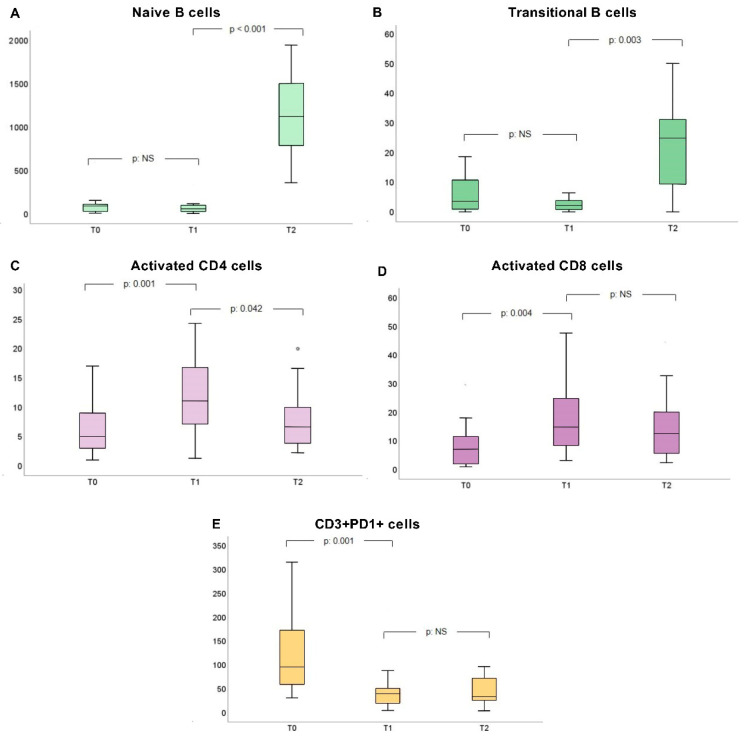
Concentrations of naïve (**A**) and transitional B-cells (**B**), activated CD4+ (**C**), activated CD8+ (**D**) and CD3+PD1+ T-cells I at T_0_, T_1_ and T_2_ in the responder group. The central tendency measures used are median value (interquartile range) (MED. (IQR)). For each of the depicted cellular subpopulations, (**A**): 77 (77) vs. 57.55 (66) vs. 1149.3 (680) cells/μL, *p*_1_: NS, *p*_2_ < 0.001; (**B**): 9 (17) vs. 1.4 (3) vs. 17.5 (21) cells/μL, *p*_1_: NS, *p*_2_: 0.003; (**C**): 6.5 (4) vs. 10.08 (11) vs. 5.29 (5) cells/μL, *p*_1_: 0.001, *p*_2_: 0.042; (**D**): 8 (19) vs. 14.76 (16) vs. 13.95 (24) cells/μL, *p*_1_: 0.004, *p*_2_: NS aI; (**E**): 130 (121) vs. 30.44 (25) vs. 27.05 (34) cells/μL, *p*_1_: 0.001, *p*_2_: NS. Abbreviations: NS: non-significant.

**Figure 3 vaccines-11-01583-f003:**
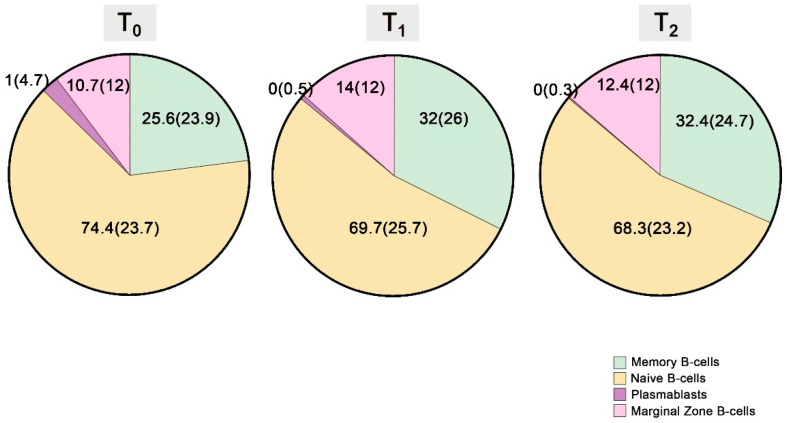
Changes in the proportions of peripheral B-lymphocyte subpopulations from T_0_ to T_2._.The central tendency measures used are median value (interquartile range) (MED (IQR)).

**Figure 4 vaccines-11-01583-f004:**
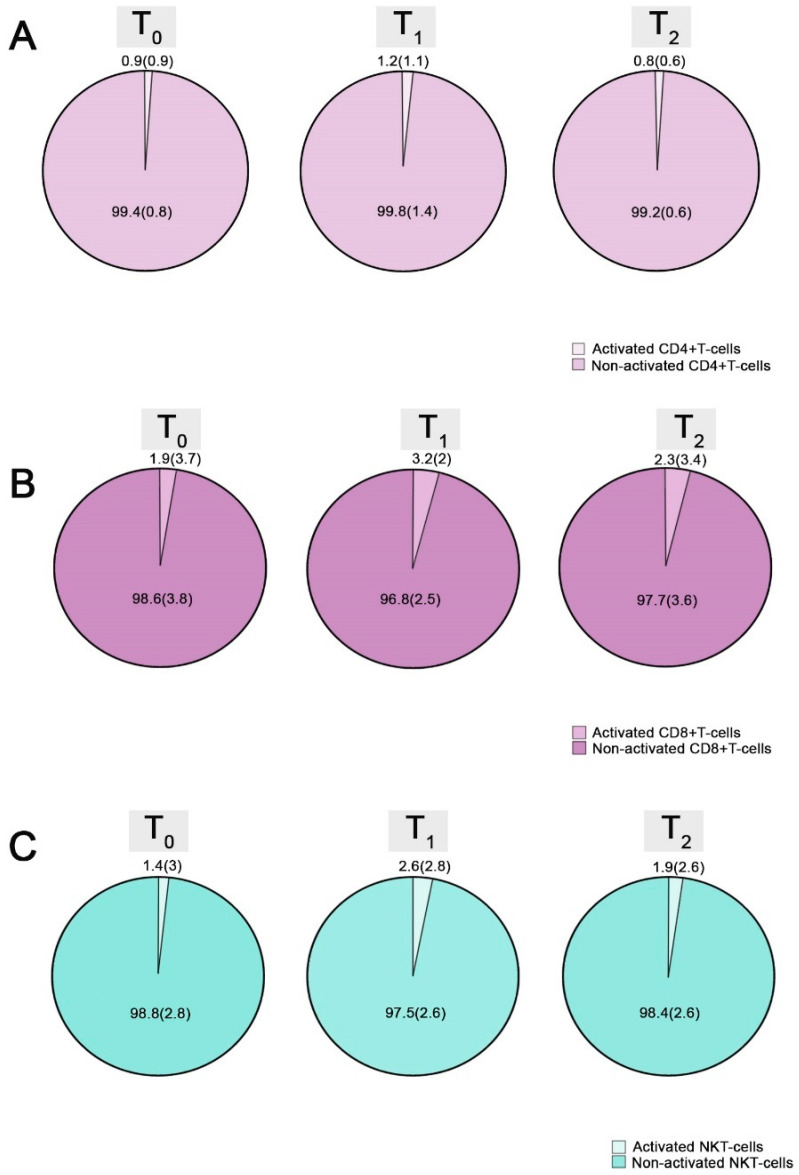
Alterations in the percentages of activated CD4+ (**A**), activated CD8+ (**B**) and activated NKT-cells (**C**) from T_0_ to T_2_: The whole pie depicts the total pool of the selected cellular subpopulation, while each piece represents the percentage of its activated or non-activated forms. The central tendency measures used are median value (interquartile range) (MED (IQR)).

**Table 1 vaccines-11-01583-t001:** Presentation of detected cell surface molecules, utilized fluorochromes and clone numbers.

CD Markers	Fluorochrome	Clone
**B-cell markers**		
CD19	PC7	J4.119
IgD	FITC	IA6-2
CD27	ECD	IA4CD27
CD38	A750	LS198-4-3
**T-cell markers**		
CD3	FITC	UCHT1
CD45	KROME	J33
CD4	APC	13B8.2
CD8	PC5.5	B9.11
CD56	A700	N901NKH-1
HLA-DR	PB	Immu-357
CD38	A750	LS198-4-3

**Table 2 vaccines-11-01583-t002:** Comparison of clinical parameters at baseline (i.e., age, eGFR, calcineurin inhibitor levels, and dialysis and transplantation vintage) between patients classified as responders or non-responders at T_2_.

Clinical Parameter	Responders	Non-Responders	*p*
Age (years)	45.5 (16)	54 (26)	NS
eGFR (mL/min/1.73 m^2^)	54.95 (24.3)	52.2 (27.4)	NS
Cyclosporin levels (ng/mL)	253.5 (218)	-	NS
Tacrolimus levels (ng/mL)	6 (2)	4.5 (2.5)	NS
Dialysis Vintage (months)	12.5 (44)	41 (30)	NS
Transplantation Vintage (years)	7.2 (14.6)	6.9 (13.5)	NS

Abbreviations: NS: non-significant. The central tendency measures used are median value (interquartile range) (MED (IQR)).

**Table 3 vaccines-11-01583-t003:** Differences in immune cell profile prior to BNT162b2 vaccination (T_0_) between patients classified as responders or non-responders at T_2_.

Cell Subpopulation Concentration (Cells/μL)	Responders	Non-Responders	*p*
WBC	7450 (2725)	7400 (3600)	NS
Lymphocytes	1772.5 (1444)	1413 (879)	NS
B-lymphocytes	96.5 (93)	51 (52)	0.045
Memory B-cells	16.5 (28)	9 (22)	NS
Naïve B-cells	77 (77)	30 (37)	NS
Transitional B-cells	9 (17)	1 (2)	0.031
Marginal Zone B-cells	5.3 (12.2)	1.7 (−)	NS
CD3+ T-cells	1399 (1325)	1076 (790)	NS
CD3+CD4+ T-cells	891 (602)	589 (456)	NS
Activated CD4+ T-cells	6.5 (4)	8 (6)	NS
CD3+CD8+ T-cells	454 (628)	389 (353)	NS
Activated CD8+ T-cells	8 (19)	7 (28)	NS
CD3+PD1+ T-cells	130 (121)	59 (81)	NS
Monocytes	173 (158)	87 (381)	NS
Activated monocytes	2 (2)	1 (11)	NS
CD3-CD56+ natural killer cells	411 (358)	487 (516)	NS
Activated natural killer T-cells	260.5 (232)	329 (384)	NS

Abbreviations: NS: non-significant. The central tendency measures used are median value (interquartile range) (MED (IQR)).

**Table 4 vaccines-11-01583-t004:** Concentrations of cell subpopulations at T_0_, T_1_ and T_2_ in the responder group.

Cell Subpopulation Concentration (Cells/μL)	T_0_	T_1_	T_2_	*p* (Friedman Test)	p_1_ (T_0_–T_1_) *	p_2_ (T_1_–T_2_) *
WBC	7450 (2725)	7750 (1725)	9050 (3200)	0.043	NS	NS
Lymphocytes	1772.5 (1444)	1790.95 (807)	1563.55 (1039)	NS	NS	NS
B-lymphocytes	96.5 (93)	84.18 (74)	83.09 (124)	NS	NS	NS
Memory B-cells	16.5 (28)	24.78 (34)	25.18 (27)	NS	NS	NS
Naïve B-cells	77 (77)	57.55 (66)	1149.3 (680)	<0.001	NS	<0.001
Transitional B-cells	9 (17)	1.4 (3)	17.5 (21)	0.002	NS	0.003
Marginal Zone B-cells	5.3 (12)	10.04 (17)	10.6 (12)	NS	NS	NS
CD3+ T-cells	1399.5 (1325)	1497.69 (829)	1375.81 (861)	NS	NS	NS
CD3+CD4+ T-cells	891 (602)	885.07 (456)	785.6 (499)	NS	NS	NS
Activated CD4+ T-cells	6.5 (4)	10.08 (11)	5.29 (5)	0.001	0.001	0.042
CD3+CD8+ T-cells	454.5 (628)	514.77 (362)	519.76 (480)	NS	NS	NS
Activated CD8+ T-cells	8 (19)	14.76 (16)	13.95 (24)	0.005	0.004	NS
CD3+PD1+ T-cells	130 (121)	30.44 (25)	27.05 (34)	<0.001	0.001	NS
Monocytes	411 (358)	389.5 (202)	422.7 (269)	NS	NS	NS
Activated monocytes	260.5 (232)	292.88 (182)	315.73 (187)	NS	NS	NS
CD3−CD56+ natural killer cells	173 (158)	124.56 (115)	160.11 (102)	NS	NS	NS
Activated natural killer T-cells	2 (2)	2.85 (4)	2.78 (3)	NS	NS	NS

* *p* values were adjusted with the use of Bonferroni correction. Abbreviations: NS: non-significant. The central tendency measures used are median value (interquartile range) (MED (IQR)).

**Table 5 vaccines-11-01583-t005:** Concentrations of cell subpopulations at T_0_, T_1_ and T_2_ in the non-responder group

Cell Subpopulation Concentration (Cells/μL)	T_0_	T_1_	T_2_	*p* (Friedman Test)	p_1_ (T_0_–T_1_) *	p_2_ (T_1_–T_2_) *
WBC	7400 (3600)	8100 (1100)	7900 (3100)	NS	NS	NS
Lymphocytes	1413 (879)	1101.6 (892)	1554 (1270)	0.041	NS	0.034
B-lymphocytes	51 (52)	39.61 (45)	26.86 (42)	NS	NS	NS
Memory B-cells	9 (22)	13.86 (18)	10.57 (19)	NS	NS	NS
Naïve B-cells	30 (37)	16.87 (32)	867.01 (1279)	0.039	NS	0.04
Transitional B-cells	1 (2)	0.34 (1)	16.17 (15)	0.039	NS	0.04
Marginal Zone B-cells	1.7 (1)	6.77 (9)	1.12 (0.9)	NS	NS	NS
CD3+ T-cells	1076 (790)	844.17 (931)	1098.6 (1466)	NS	NS	NS
CD3+CD4+ T-cells	589 (456)	492.25 (625)	642.21 (1014)	NS	NS	NS
Activated CD4+ T-cells	8 (6)	5.97 (11)	5.31 (7)	NS	NS	NS
CD3+CD8+ T-cells	389 (353)	359.85 (320)	458.93 (443)	NS	NS	NS
Activated CD8+ T-cells	7 (28)	10.5 (21)	8.29 (14)	NS	NS	NS
CD3+PD1+ T-cells	59 (81)	32.82 (40)	59.85 (72)	NS	NS	NS
Monocytes	487 (516)	495.9 (162)	394.6 (285)	NS	NS	NS
Activated monocytes	329 (384)	344.65 (180)	240.9 (241)	NS	NS	NS
CD3-CD56+ natural killer cells	87 (381)	74.91 (164)	192.75 (378)	0.05	NS	NS
Activated natural killer T-cells	1 (11)	2.17 (4)	1.56 (3.1)	NS	NS	NS

* *p* values were adjusted with the use of Bonferroni correction. Abbreviations: NS: non-significant. The central tendency measures used are median value (interquartile range) (MED (IQR)).

**Table 6 vaccines-11-01583-t006:** Percentages of cell subpopulations at T_0_, T_1_ and T_2_ in the responder group.

Cell Subpopulation Percentage (%)	T_0_	T_1_	T_2_	*p* (Friedman Test)	p_1_ (T_0_–T_1_) *	p_2_ (T_1_–T_2_) *
Lymphocytes	23 (18.1)	21.7 (10.3)	21.3 (5.8)	0.014	0.027	NS
B-lymphocytes	4.4 (5)	5.5 (3.7)	6.2 (4.6)	NS	NS	NS
Memory B-cells	25.6 (23.9)	32 (26)	32.4 (24.7)	0.001	0.001	NS
Naïve B-cells	74.4 (23.7)	69.7 (25.7)	68.3 (23.2)	0.001	0.002	NS
Plasmablasts	1 (4.7)	0 (0.5)	0 (0.3)	0.03	NS	NS
Switched plasmablasts	19 (66.7)	0 (100)	0 (100)	NS	NS	NS
Non-switched plasmablasts	0 (58.3)	0 (0)	0 (0)	NS	NS	NS
Transitional B-cells	9.4 (30.6)	2 (4.9)	1.8 (2.1)	0.006	NS	NS
Marginal Zone B-cells	10.7 (12.4)	14 (12.4)	12.4 (12.2)	0.016	0.031	NS
CD3+ T-cells	83.3 (15.3)	82.5 (8.9)	81.8 (7.9)	NS	NS	NS
CD3+CD4+ T-cells	50.5 (15.4)	49.5 (11.8)	49.8 (13.6)	NS	NS	NS
Activated CD4+ T-cells	0.9 (0.9)	1.2 (1.1)	0.8 (0.6)	0.001	0.001	0.031
CD3+CD8+ T-cells	30 (17)	30.4 (12)	30.7 (13.9)	0.002	NS	0.002
Activated CD8+ T-cells	1.9 (3.7)	3.2 (2)	2.3 (3.4)	0.02	0.017	NS
CD3+PD1+ T-cells	7.3 (4.7)	1.8 (1.4)	2.2 (2.1)	<0.001	<0.001	NS
Monocytes	5 (3.7)	4.8 (3)	5.1 (2.4)	NS	NS	NS
Activated monocytes	77 (22.3)	77.1 (24.8)	75.4 (21.8)	NS	NS	NS
CD3-CD56+ natural killer cells	8.4 (4.9)	8.3 (5.6)	8.9 (5.7)	NS	NS	NS
Activated natural killer T-cells	1.4 (3)	2.6 (2.8)	1.9 (2.6)	0.001	0.001	0.03

* *p* values were adjusted with the use of Bonferroni correction. (a) Percentage explanation: lymphocytes and monocytes are estimated as percentages of WBCs; (b) B-lymphocytes, CD3+, CD3+PD1+, CD3+CD4+ and CD3+CD8+ T-cells and CD3-CD56+ natural killer cells as percentages of lymphocytes; (c) memory, naïve, marginal zone B-cells and plasmablasts as percentages of B-lymphocytes; (d) switched and non-switched plasmablasts as percentages of plasmablasts; (e) activated CD4+ and CD8+ T-cells as percentages of CD3+CD4+ and CD3+CD8+ T-cells, respectively; (f) activated monocytes as a percentage of monocytes; and (g) activated natural killer T-cells as a percentage of CD3-CD56+ natural killer cells. Abbreviations: NS: non-significant. The central tendency measures used are median value (interquartile range) (MED (IQR)).

**Table 7 vaccines-11-01583-t007:** Percentages of cell subpopulations at T_0_, T_1_ and T_2_ in the non-responder group.

Cell Subpopulation Percentage (%)	T_0_	T_1_	T_2_	*p* (Friedman Test)	p_1_ (T_0_–T_1_) *	p_2_ (T_1_–T_2_) *
Lymphocytes	18.5 (7.4)	13.6 (11.4)	18.5 (14.7)	NS	NS	NS
B-lymphocytes	3 (2.9)	2.8 (2.7)	1.55 (2.1)	0.05	NS	NS
Memory B-cells	27.9 (26)	32.7 (21.3)	36.95 (27.6)	NS	NS	NS
Naïve B-cells	73.5 (25.7)	69.2 (21.9)	62.2 (28)	NS	NS	NS
Plasmablasts	1.45 (4.3)	1.2 (3.6)	0.55 (3.3)	NS	NS	NS
Switched plasmablasts	100 (50)	100 (62.5)	50 (100)	NS	NS	NS
Non-switched plasmablasts	0 (0)	0 (12.5)	0 (0)	NS	NS	NS
Transitional B-cells	2 (5.5)	2 (2.3)	2.55 (3.1)	NS	NS	NS
Marginal Zone B-cells	17.6 (−)	12.3 (12.5)	16.1 (16.7)	NS	NS	NS
CD3+ T-cells	82.4 (23.6)	78 (18.8)	79 (19.2)	NS	NS	NS
CD3+CD4+ T-cells	52.4 (31.1)	51.7 (19.3)	54.2 (27.1)	NS	NS	NS
Activated CD4+ T-cells	1.4 (1.8)	0.9 (1.9)	0.65 (0.9)	NS	NS	NS
CD3+CD8+ T-cells	26.6 (13.3)	24.8 (16)	26.6 (14.8)	NS	NS	NS
Activated CD8+ T-cells	1.9 (5.7)	2.7 (4.4)	1.7 (2)	NS	NS	NS
CD3+PD1+ T-cells	4.7 (4.1)	2.9 (2.1)	3.55 (2.2)	0.039	0.04	NS
Monocytes	8.3 (5.5)	5.7 (2.4)	4.95 (2.4)	NS	NS	NS
Activated monocytes	66.9 (9.7)	70.8 (17.7)	62.85 (24.6)	NS	NS	NS
CD3-CD56+ natural killer cells	7.9 (21.9)	10.5 (15.2)	14.1 (19.7)	NS	NS	NS
Activated natural killer T-cells	1.1 (2.3)	2.9 (1.6)	0.8 (-)	NS	NS	NS

* *p* values were adjusted with the use of Bonferroni correction. (a) Percentage explanation: lymphocytes and monocytes are estimated as percentages of WBCs; (b) B-lymphocytes, CD3+, CD3+PD1+, CD3+CD4+ and CD3+CD8+ T-cells and CD3-CD56+ natural killer cells as percentages of lymphocytes; (c) memory, naïve, marginal zone B-cells and plasmablasts as percentages of B-lymphocytes; (d) switched and non-switched plasmablasts as percentages of plasmablasts; (e) activated CD4+ and CD8+ T-cells as percentages of CD3+CD4+ and CD3+CD8+ T-cells, respectively; (f) activated monocytes as a percentage of monocytes; and (g) activated natural killer T-cells as a percentage of CD3-CD56+ natural killer cells. Abbreviations: NS: non-significant. The central tendency measures used are median value (interquartile range) (MED (IQR)).

## Data Availability

Patients’ data are available on request from the authors.

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
