# Peer review of "Immune Profile Determines Response to Vaccination against COVID-19 in Kidney Transplant Recipients"

_vaccines, 2023, doi:10.3390/vaccines11101583_

Round 1

Reviewer 1 Report

Stai et al. report on a prospective monocentre study of the transplant follow up clinic at the University of Thessaloniki. 56 COVID naïve kidney transplant patients receiving standard immunosuppression and their PBMC’s were investigated. FACS analyses were performed and the changes of the different lymphocyte population correlated to antibody response. Patients with higher B cell count and more transitional B cells had a significantly better vaccination response. The manuscript is well written and I have the following comments:

Major

1.     The study was limited to stable transplant patients excluding heavier immunosuppression for pre-existing diseases, chemotherapy and recent rejection intervention. This allowed a cleaner population but restricts the generalisation of the findings. The study population was also of a smaller size (n=56 with an n=39 in the final analysis and only four non-responders). I would propose to expand on these aspects as limitations in the discussion.

2.     The background TX medication isn’t depicted and it would be interesting to see whether a certain amount of therapy associated with vaccine non-response. I would propose to add this fact to the limitation if it isn’t possible to add the immunosuppression. 

Minor

1. I would suggest to add the patient population and study design to the abstract stating that a prospective monocentre study was conducted.

2. Perhaps the authors would consider changing renal to kidney e.g. kidney transplant patients.

3. I would suggest to change the colour so to increase the distinction of the cell population in figure 3.

Author Response

Dear reviewers, 

Thank very much for you time and effort to review and comment on our manuscript. We have thoroughly revised the paper under your suggestions, which we believe were professional and helpfull and gave us the opportunity to better present our work

Comments and Suggestions for Authors

Stai et al. report on a prospective monocentre study of the transplant follow up clinic at the University of Thessaloniki. 56 COVID naïve kidney transplant patients receiving standard immunosuppression and their PBMC’s were investigated. FACS analyses were performed and the changes of the different lymphocyte population correlated to antibody response. Patients with higher B cell count and more transitional B cells had a significantly better vaccination response. The manuscript is well written and I have the following comments:

Major

  1. The study was limited to stable transplant patients excluding heavier immunosuppression for pre-existing diseases, chemotherapy and recent rejection intervention. This allowed a cleaner population but restricts the generalisation of the findings. The study population was also of a smaller size (n=56 with an n=39 in the final analysis and only four non-responders). I would propose to expand on these aspects as limitations in the discussion.

Thank you for your comment, you are absolutely right. We have added a relevant paragraph in the discussion section.

  1. The background TX medication isn’t depicted and it would be interesting to see whether a certain amount of therapy associated with vaccine non-response. I would propose to add this fact to the limitation if it isn’t possible to add the immunosuppression.

Thank you for your observation. All our patients were under a triple combination of corticosteroids, calcineurin inhibitors and mycophenolate mofetil (we added that information in the “inclusion criteria” section). However, quantification of immunosuppressive treatment could not be performed, as we cannot measure mycophenolate mofetil serum levels, and therefore, we could not estimate potential correlation between immunosuppressive treatment dosage and vaccination responsiveness, so we listed that as a study limitation.

Minor

  1. I would suggest to add the patient population and study design to the abstract stating that a prospective monocentre study was conducted.

Thank you, we have added that in the abstract.

  1. Perhaps the authors would consider changing renal to kidney e.g. kidney transplant patients.

Ok, we have corrected that.

  1. I would suggest to change the colour so to increase the distinction of the cell population in figure 3.

Thank you, we have changed that.

Reviewer 2 Report

1.There are some writing and formatting errors in the text, please check and correct them carefully.

2. In line 42 of the article, what is the basis for RTRs being prone to complications?Please cite references for description.

3. The author has separately introduced the background of vaccines and the background of RTRs, so why study the relationship between these two? What is the basis? What is the actual problem to be solved? Please elaborate in the introduction.

4, Line 65 of the article, RTRs immunity has not been extensively studied, so to study the relationship between SARS-CoV-2 vaccine and RTRs immunity? I think it's too far-fetched to convince the reader. Please put it another way or give a fuller explanation.

5. Please indicate the number and approval unit of the informed consent form in the method section.

6. All the tables in the results section of the article are not annotated, and what do the numbers in parentheses mean? There is no clear statement in the article, please revise one by one.

7. What does line 210 of the passage, 34/39, mean? 34 of 39 participants? If I understand you correctly, I suggest using words instead of numbers.

8. The figures in the article also have the same problem as the table. Please mark the meaning of the numbers in parentheses. The images also lack annotations.

9. In the discussion, some complicated and unnecessary parts can be condensed, especially the first few paragraphs.

10. It is suggested to sort out the logic of part of the discussion, discuss the significance of which part of the indicators, and demonstrate before and after it. The logic of the discussion part is not clear.

11. In the article, if the conclusion is not drawn through your own research, please mark the references and check carefully whether all the references are cited? Especially the discussion section.

12. I think it would be better if the conclusions were written separately from the limitations of the study. Authors may consider putting the limitations of the study at the end of the discussion section of the article or writing a separate section.

Author Response

Dear reviewers, 

thank very much for you time and effort to review and comment on our manuscript. We have thoroughly revised the paper under your suggestions, which we believe were professional and helpfull and gave us the opportunity to better present our work

1.There are some writing and formatting errors in the text, please check and correct them carefully.

Thank you for your comment, we have tried to correct as many of these errors as possible and hope the text has significantly improved.

  1. In line 42 of the article, what is the basis for RTRs being prone to complications? Please cite references for description.

Thank you for your comment. We have now modified the text in order to make clear that RTRs are more prone to COVID-19 complications due to the fact that they are immunocompromised and also present with several comorbidities. We have also added the relevant references.

  1. The author has separately introduced the background of vaccines and the background of RTRs, so why study the relationship between these two? What is the basis? What is the actual problem to be solved? Please elaborate in the introduction.

That is a correct comment. Our aim was to analyze the effect BNT162b2 vaccination has on RTRs’ immune system and vice versa (since they comprise a distinct group of severely immunocompromised patients and BNT162b2 is the most widely used vaccine among them). The results of this study could potentially be helpful in the design of future vaccination strategies with regards to RTRs, not only against the SARS-CoV-2, but also other pathogens. We have modified the content of the introduction section and hope our point has now been made clearer.

4, Line 65 of the article, RTRs immunity has not been extensively studied, so to study the relationship between SARS-CoV-2 vaccine and RTRs immunity? I think it's too far-fetched to convince the reader. Please put it another way or give a fuller explanation.

Thank you, we had possibly not managed to put that correctly… RTRs’ immunity has already been extensively studied; however, we lack data regarding its interaction with anti-SARS-CoV-2 vaccines. We hope our point has now been sufficiently elucidated.

  1. Please indicate the number and approval unit of the informed consent form in the method section.

Thank you very much for this comment, we have added these data

  1. All the tables in the results section of the article are not annotated, and what do the numbers in parentheses mean? There is no clear statement in the article, please revise one by one.

Thank you very much for this suggestion, we have done that and revised all tables in the manuscript.

  1. What does line 210 of the passage, 34/39, mean? 34 of 39 participants? If I understand you correctly, I suggest using words instead of numbers.

Thank you, we have corrected that.

  1. The figures in the article also have the same problem as the table. Please mark the meaning of the numbers in parentheses. The images also lack annotations.

Thank you, we have added the relevant information.

  1. In the discussion, some complicated and unnecessary parts can be condensed, especially the first few paragraphs.

You are absolutely right; a considerable part of the discussion section could be considered quite confusing. In the corrected text, we have attempted to skip a few unnecessary parts and reformulate several sentences that needed improvement.

  1. It is suggested to sort out the logic of part of the discussion, discuss the significance of which part of the indicators, and demonstrate before and after it. The logic of the discussion part is not clear.

Thank you for your comment. We have modified some parts, aiming to make the structure of the discussion section easier to follow: more specifically, we initially present each of our findings and then compare it to these of other researchers, attempting to interpret the underlying mechanisms.  

  1. In the article, if the conclusion is not drawn through your own research, please mark the references and check carefully whether all the references are cited? Especially the discussion section.

Thank you, we have taken care of that and have added any missing references.

  1. I think it would be better if the conclusions were written separately from the limitations of the study. Authors may consider putting the limitations of the study at the end of the discussion section of the article or writing a separate section.

Thank you, the limitations of our study are now presented at the end of the discussion part.

Reviewer 3 Report

The manuscript "Immune profile determines response to vaccination against 2 Covid-19 in renal transplant recipients" by Stai et al is a nice, impressive work. The study design is impressive. The methodology, results and discussion are well elaborated and in accordance with each other. There are few minor points from my side. The inclusion/exclusion is well planned and elaborated as well.

1. I could not see any ethical approval. Does this work not need any ethical approval?

2. The authors are advised to check for minor English and grammar errors.

3. The authors are advised to add a strong future perspective mentioning their own opinion on how the findings of their study will help/aid researchers in immediate future.

Minor editing needed

Author Response

Dear reviewers, 

thank very much for you time and effort to review and comment on our manuscript. We have thoroughly revised the paper under your suggestions, which we believe were professional and helpfull and gave us the opportunity to better present our work

  1. I could not see any ethical approval. Does this work not need any ethical approval?

Thank you very much for this comment, by mistake, we had put the ethics approval. We added now, in the Methods section

  1. The authors are advised to check for minor English and grammar errors.

Thank you, we have tried to improve the text.

  1. The authors are advised to add a strong future perspective mentioning their own opinion on how the findings of their study will help/aid researchers in immediate future.

You are absolutely right. We have done that.

Comments on the Quality of English Language

Minor editing needed

Thank you very much, we have edited the whole manuscript